# Interpregnancy interval and adverse pregnancy outcomes among pregnancies following miscarriages or induced abortions in Norway (2008–2016): A cohort study

Gizachew A. Tessema[1,2,3]*, Siri E. Håberg[3], Gavin Pereira[1,3], Annette K. Regan[1,4,5], Jennifer Dunne[1], Maria C. Magnus[3]

1 Curtin School of Population Health, Curtin University, Perth, Western Australia, Australia, 2 School of Public Health, University of Adelaide, Adelaide, South Australia, Australia, 3 Centre for Fertility and Health, Norwegian Institute of Public Health, Oslo, Norway, 4 School of Nursing and Health Professions, University of San Francisco, Orange, California, United States of America, 5 Fielding School of Public Health, University of California Los Angeles, Los Angeles, California, United States of America

* gizachew.tessema@curtin.edu.au

**Data Availability Statement:** Study data are available on application. Information on how to

## Abstract

### Background

The World Health Organization recommends to wait at least 6 months after miscarriage and induced abortion before becoming pregnant again to avoid complications in the next pregnancy, although the evidence-based underlying this recommendation is scarce. We aimed to investigate the risk of adverse pregnancy outcomes—preterm birth (PTB), spontaneous PTB, small for gestational age (SGA) birth, large for gestational age (LGA) birth, preeclampsia, and gestational diabetes mellitus (GDM)—by interpregnancy interval (IPI) for births following a previous miscarriage or induced abortion.

### Methods and findings

We conducted a cohort study using a total of 49,058 births following a previous miscarriage and 23,707 births following a previous induced abortion in Norway between 2008 and 2016. We modeled the relationship between IPI and 6 adverse pregnancy outcomes separately for births after miscarriages and births after induced abortions. We used log-binomial regression to estimate unadjusted and adjusted relative risk (aRR) and 95% confidence intervals (CIs). In the adjusted model, we included maternal age, gravidity, and year of birth measured at the time of the index (after interval) births. In a sensitivity analysis, we further adjusted for smoking during pregnancy and prepregnancy body mass index. Compared to births with an IPI of 6 to 11 months after miscarriages (10.1%), there were lower risks of SGA births among births with an IPI of <3 months (8.6%) (aRR 0.85, 95% CI: 0.79, 0.92, $p < 0.01$) and 3 to 5 months (9.0%) (aRR 0.90, 95% CI: 0.83, 0.97, $p = 0.01$). An IPI of <3 months after a miscarriage (3.3%) was also associated with lower risk of GDM (aRR 0.84, 95% CI: 0.75, 0.96, $p = 0.01$) as compared to an IPI of 6 to 11 months (4.5%). For births following an induced abortion, an IPI <3 months (11.5%) was associated with a nonsignificant

apply and access data can be found from https://helsedata.no/. Data access is subject to compulsory ethics and governance approvals.

**Funding:** This work was supported with funding from the Research Council of Norway through its Centres of Excellence funding scheme (#262700 to GAT, SEH, GP, and MCM, and #320656 to SEH and MCM). GAT is funded by the National Health and Medical Research Council Investigator Grant (#1195716) and the Charter Hall Collaborative Grant from the Raine Medical Research Foundation (#RCA02-20). GP is funded by the National Health and Medical Research Council Project (#1099655) and Investigator Grants (#1173991). MCM is funded by the European Research Council (ERC) under the European Union's Horizon 2020 Research and Innovation Programme (#947684). The funders had no role in study design, data collection and analysis, decision to publish, or preparation of the manuscript.

**Competing interests:** The authors have declared that no competing interests exist.

**Abbreviations:** aRR, adjusted relative risk; BMI, body mass index; CI, confidence interval; GDM, gestational diabetes mellitus; HELLP, haemolysis, elevated liver enzymes, and low platelet count; ICD, International Classification of Diseases; IPI, interpregnancy interval; IQR, interquartile range; LBW, low birth weight; LGA, large for gestational age; OR, odds ratio; PTB, preterm birth; SGA, small for gestational age; WHO, World Health Organization.

but increased risk of SGA (aRR 1.16, 95% CI: 0.99, 1.36, $p = 0.07$) as compared to an IPI of 6 to 11 months (10.0%), while the risk of LGA was lower among those with an IPI 3 to 5 months (8.0%) (aRR 0.84, 95% CI: 0.72, 0.98, $p = 0.03$) compared to an IPI of 6 to 11 months (9.4%). There was no observed association between adverse pregnancy outcomes with an IPI >12 months after either a miscarriage or induced abortion ($p > 0.05$), with the exception of an increased risk of GDM among women with an IPI of 12 to 17 months (5.8%) (aRR 1.20, 95% CI: 1.02, 1.40, $p = 0.02$), 18 to 23 months (6.2%) (aRR 1.24, 95% CI: 1.02, 1.50, $p = 0.03$), and $\geq$24 months (6.4%) (aRR 1.14, 95% CI: 0.97, 1.34, $p = 0.10$) compared to an IPI of 6 to 11 months (4.5%) after a miscarriage. Inherent to retrospective registry-based studies, we did not have information on potential confounders such as pregnancy intention and health-seeking bahaviour. Furthermore, we only had information on miscarriages that resulted in contact with the healthcare system.

## Conclusions

Our study suggests that conceiving within 3 months after a miscarriage or an induced abortion is not associated with increased risks of adverse pregnancy outcomes. In combination with previous research, these results suggest that women could attempt pregnancy soon after a previous miscarriage or induced abortion without increasing perinatal health risks.

## Author summary

### Why was this study done?

- The World Health Organization recommends to wait at least 6 months after miscarriage and induced abortion before becoming pregnant again to avoid complications in the next pregnancy, although the evidence-based underlying this recommendation is scarce.

- The differences in pregnancy outcomes according to interpregnancy interval (IPI) after miscarriage as opposed to induced abortions remains unclear.

### What did the researchers do and find?

- Using data linkage from registry data in Norway, we explored the risks of adverse pregnancy outcomes for births after a miscarriage and after an induced abortion separately. While 3 out of 5 women with previous miscarriages conceive within 6 months, 1 out of 5 women with previous induced abortions conceive within 6 months.

- Our study suggests that conceiving within 6 months after a miscarriage or an induced abortion is not associated with increased risks of adverse pregnancy outcomes. The results are consistent for IPI as short as 3 months.

- There was no evidence of higher risks of adverse pregnancy outcomes among women with an IPI of greater than 12 months after a miscarriages or induced abortions, with the exception of a modest increased risk of gestational diabetes mellitus.

**What do these findings mean?**

- Our results do not support current international recommendations to wait at least 6 months after a miscarriage or an induced abortion.

- In combination with previous research, our results are reassuring for families who attempt pregnancy soon after a miscarriage or induced abortion.

- These findings motivate a review of current international guidelines for birth spacing after a miscarriage or an induced abortion.

## Introduction

Miscarriage or the loss of the fetus before viability occurs in 12% to 15% of recognised pregnancies [1–3], and induced abortions occur in up to 15% of pregnancies in high-income countries [4]. Miscarriage causes significant psychological distress for couples [5], and induced abortions are performed for many different reasons including medical, financial, or social reasons [6,7]. The optimal interpregnancy interval (IPI)—the time between the end of one pregnancy and the start of the next—after pregnancy loss remains unclear. Based on a single study from Latin America, which reported that an IPI less than 6 months following miscarriages and induced abortions was associated with increased risk of adverse perinatal outcomes in the next pregnancy (odds ratio (OR) > 2.0 for preterm birth (PTB) and low birth weight (LBW)) [8], the World Health Organization (WHO) has recommended since 2007 that women wait at least 6 months before trying to become pregnant again after a miscarriage or induced abortion [9]. The study from Latin America did not distinguish between miscarriages and induced abortions, which is likely to have resulted in substantial heterogeneity in the underlying risk profile. Furthermore, when large cohort studies from Scotland (OR 0.89, 95% confidence interval (CI): 0.81, 0.98 for PTB; OR 0.84, 95% CI: 0.71, 0.89 for LBW; and OR 1.18, 95% CI: 0.82, 1.23 for preeclampsia) and California (OR 0.87, 95% CI: 0.81, 0.94 for PTB) subsequently evaluated this research question, they observed no increased risk of adverse pregnancy outcomes in births that followed a short IPI (<6 months) after a miscarriage [10,11]. However, an increased risk of adverse pregnancy outcomes in pregnancies that followed a short (<6 months) IPI after induced abortions was observed in Finland (OR 1.44, 95% CI: 1.10, 1.88 for PTB) [12]. Studies investigating risks of maternal complications after long IPI following a live birth indicated increased risks of gestational diabetes mellitus (GDM) in Canada (OR 1.66, 95% CI: 1.39, 2.00) and preeclampsia in Australia (OR 1.28, 95% CI: 1.17, 1.41) [13,14]. The differences in pregnancy outcomes according to IPI after miscarriage as opposed to induced abortions therefore remains unclear. Besides the available large cohort studies described above, other studies with smaller sample size ($n < 1,100$) were conducted to explore pregnancy outcomes after a miscarriage; however, these studies investigated limited outcomes such as recurrent miscarriage and PTB [15–18]. Therefore, there is a dearth of research exploring the subsequent risks of other pregnancy outcomes including preeclampsia and GDM following miscarriage or induced abortion. Our objective was therefore to investigate the risk of a broad range of adverse pregnancy outcomes such as PTB, spontaneous PTB, small for gestational age (SGA), large for gestational age (LGA), preeclampsia, and GDM according to IPI for pregnancies that follow miscarriages or induced abortions separately, with the intention of informing intrapartum care following early pregnancy loss.

## Methods

### Study design and data sources

We conducted a retrospective cohort study using 3 Norwegian national health registries: the Medical Birth Registry of Norway (birth registry) [19], the Norwegian Patient Registry (patient registry), and the General Practitioner database [20]. Using these 3 registries, we identified all registered pregnancies in Norway with an estimated date of conception between January 1, 2008 and December 31, 2016. The birth registry includes mandatory notifications on pregnancies in Norway ending after 12 gestational weeks and provided information on live births, stillbirths, miscarriages, and late induced abortions.

The patient and general practitioners' registries provided information on induced abortions and miscarriages irrespective of gestational weeks. A detailed description of the processes for identification of miscarriage and induced abortions and data linkage procedures have been described previously [21]. In summary, we used primary and secondary diagnostic codes indicating the presence of a miscarriage or induced abortions. From the patient registry, information on miscarriages before 12 completed gestational weeks were captured using the following International Classification of Diseases version 10 (ICD-10) codes: missed abortion (O02.1); other specified abnormal products of conception (O02.8); abnormal product of conception, unspecified (O02.9); spontaneous abortion (O03); and threatened abortion (O20.0). Similarly, induced abortions were identified using the following ICD-10 codes: medical abortion (O04), other abortion (O05), and unspecified abortion (O06). In the general practitioner database, we captured information on miscarriages before 12 completed gestational weeks using the following ICPC-2 codes: bleeding in pregnancy (W03) and spontaneous abortion (W82). We only counted registrations of threatened abortion and bleeding in pregnancy codes as miscarriage or induced abortions if they did not subsequently end in a registration in the birth registry.

In this study, a fetal death at 20 gestational weeks or later or with a birthweight of 400 grams or more was considered a stillbirth, while fetal deaths prior to 20 gestational weeks with a birthweight less than 400 grams were defined as miscarriages. This study is reported as per the Strengthening the Reporting of Observational Studies in Epidemiology (STROBE) guideline [22] (S1 STROBE Checklist).

### Participants and exclusions

In the study period, there were a total of 108,444 miscarriages of which 75,059 were followed by subsequent pregnancies, and 127,912 induced abortions of which 57,282 were followed by subsequent pregnancies. After excluding pregnancies ended prior to 20 weeks, we were left with 50,343 births after miscarriages and 24,248 births after induced abortions. By further excluding multiple births, births with gestational age $\geq$45 weeks, late induced abortions, births with missing maternal age, and births with missing information on birthweight or offspring sex, we had a sample size of 49,058 index births after previous miscarriage and 23,707 index births after previous induced abortions. The index birth was defined as the birth after the IPI. A total of 1,647 (3.4%) women after miscarriages and 552 (2.3%) women after induced abortions contributed more than one birth in the study population (Fig 1).

### Exposure

IPI was defined as the time between the date of previous miscarriage or induced abortion and date of the conception (date of birth minus gestational age) of the subsequent live or stillbirth recorded in the birth registry. The gestational age estimation was largely based on ultrasound estimates (99%). We categorised IPI into 6 categories: <3, 3 to 5, 6 to 11, 12 to 17, 18 to 23,

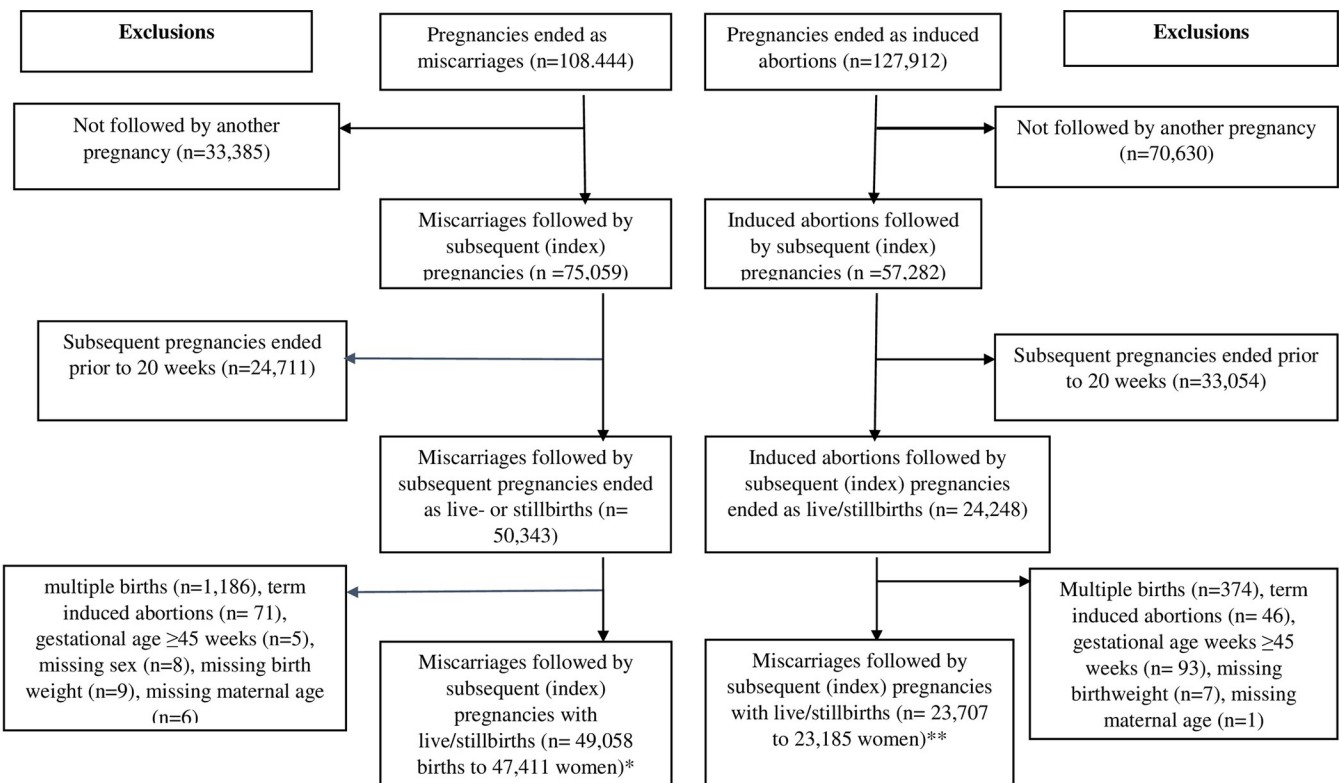

*N=1,647 women contributed more than one pregnancy in the cohort. **n=552 women contributed more than one pregnancy in the cohort

**Fig 1. Flow chart for selection of births following miscarriages and induced abortions between 2008 and 2016 in Norway.**

and ≥24 months, with 6 to 11 months as the reference category, consistent with the WHO guidelines and literature [9,10,23].

## Outcomes

We studied 6 adverse pregnancy outcomes: PTB, spontaneous PTB, SGA, LGA, preeclampsia, and GDM. [10,24–26] PTB was defined as birth occurring before 37 completed weeks of gestation. Spontaneous PTB was defined as PTB with spontaneous onset of labour. SGA and LGA were defined as a birthweight in the lowest or highest 10th percentiles, based on the gestational week and sex-specific distributions of birthweight among all births in the birth registry. Preeclampsia was defined as any registration of preeclampsia, eclampsia, or HELLP syndrome (haemolysis, elevated liver enzymes, and low platelet count). GDM was defined based on the WHO-1999 criteria when fasting plasma glucose level ≥7.0 mmol/L or glucose tolerance test of ≥7.8 mmol/L and <11.0 mmol/l [27].

## Statistical analysis

We evaluated the relationship between IPI and the pregnancy outcomes separately for births after miscarriages and births after induced abortions. We used log-binomial regression to estimate unadjusted and adjusted relative risk (aRR) and 95% CIs [17]. Robust cluster variance estimation was used to account for women who had more than one pregnancy during the study period. We adjusted for maternal age using restricted cubic splines with 5 knots (placed at the 5th, 27.5th, 50th, 72.5th, and 95th percentiles in the study population; [28]), gravidity

(categorical: 1, 2, 3, or more), and year of birth (continuous) at the time of index birth. We tested for the multiplicative interaction between IPI and maternal age, and the interaction between IPI and parity, using likelihood ratio test comparing models with and without interaction terms. We calculated *e*-values for the aRR as a measure of the minimum strength of association a confounder would have to have with the exposure and outcome to explain away the observed associations. Small *e*-values indicates that little unmeasured confounding is required to explain or nullify observed effects. The converse is true of high *e*-values. The *e*-value for the lower limit of the 95% CI represents the level of confounding required to render the interval estimate null [29].

We conducted sensitivity analyses additionally adjusting for prepregnancy body mass index (BMI) (using restricted cubic splines placed at 5 knots placed at the 5th, 27.5th, 50th, 72.5th, and 95th percentiles in the study population) and smoking during pregnancy (categorical: yes/ no) for the index pregnancies with information available ($n = 27,747$ for births after a miscarriage; $n = 13,932$ for births after an induced abortion). We also explored adjustment for characteristics of the miscarriage or induced abortion prior to the index pregnancy (start of the IPI), such as maternal age, gravidity, and year of end of pregnancy [30]. Since IPI categorisation that consider <6 months in the short IPI category is widely considered in the literature and for our results to be compared with the WHO recommendation, we conducted additional analysis combining the first 2 IPI categories (<3 months and 3 to 5 months). We also conducted a sensitivity analysis restricting births from women with only 1 miscarriage ($n = 47,411$) or induced abortion ($n = 23,185$) in the cohort. All analyses were conducted using STATA version 16 (Statacorp, Texas).

## Ethics statement

Ethics approval was obtained the Regional Committee for Medical and Health Research Ethics of South/East Norway (2014/404).

## Results

### Maternal characteristics at the time of birth after miscarriages and induced abortions

We included 49,058 index births after miscarriage and 23,707 index births after an induced abortion. The median maternal age at the index birth was 29 years (interquartile range (IQR) = 25 to 33 years) among women with births after a miscarriage and 28 years (IQR = 24 to 48) among women with births after an induced abortion (Tables 1 and 2).

### Previous miscarriage

The median IPI after a miscarriage was 4 months (IQR = 2 to 9 months); 61% ($n = 30,076$) conceived again within 6 months and 20% ($n = 9,975$) conceived between 6 and 11 months (Table 1 and Fig 2). In births following a previous miscarriage, there were 6.0% PTB, 3.5% spontaneous PTB, 9.5% SGA births, 10.3% LGA births, 3.2% preeclampsia, and 4.2% GDM (S1 Table).

In the adjusted analysis, women with an IPI <3 months (8.6%) (aRR 0.85, 95% CI: 0.79, 0.92, $p < 0.01$) and 3 to 5 months (9.0%) (aRR 0.89, 95% CI: 0.82, 0.96, $p = 0.01$) had a lower risk of SGA compared to women with an IPI of 6 to 11 months (10.1%). We also observed a lower risk of GDM (aRR 0.85, 95% CI: 0.75, 0.95, $p = 0.01$) among women with an IPI of <3 months (3.3%) compared to women with an IPI of 6 to 11 months (4.5%).

**Table 1. Characteristics for women with births after miscarriage and induced abortion at the time of birth after the IPI for births between 2008 and 2016 in Norway (*n* = 49,058).**

| Variable | Number (%) |
|---|---|
| **Maternal age (in years)** | |
| 14–19 | 377 (0.8) |
| 20–24 | 5,148 (10.5) |
| 25–29 | 13,157 (26.8) |
| 30–34 | 16,174 (33.0) |
| 35–39 | 10,867 (22.1) |
| ≥40 | 3,335 (6.8) |
| Median (IQR) | 29 (25–33) |
| **Gravidity** | |
| 1 | 19,602 (40.1) |
| 2 | 14,305 (29.2) |
| 3+ | 15,151 (30.8) |
| **Prepregnancy BMI (kg/m$^2$)** | |
| <18.5 | 986 (2.0) |
| 18.5–25 | 17,611 (35.9) |
| 25–30 | 7,213 (14.7) |
| ≥30 | 4,123 (8.4) |
| Missing | 19,127 (39.0) |
| **Smoking during pregnancy** | |
| No | 39,302 (80.1) |
| Yes | 3,780 (7.7) |
| Missing | 5,976 (12.2) |
| **Birth year** | |
| 2008 | 238 (0.5) |
| 2009 | 4,358 (8.9) |
| 2010 | 5,955 (12.1) |
| 2011 | 6,241 (12.7) |
| 2012 | 6,330 (12.9) |
| 2013 | 6,364 (12.9) |
| 2014 | 6,364 (13.0) |
| 2015 | 6,607 (13.5) |
| 2016 | 6,585 (13.4) |
| **IPI in months** | |
| <3 | 17,251 (35.2) |
| <6 | 30,076 (61.3) |
| 6–11 | 9,975 (20.3) |
| 12–17 | 3,653 (7.5) |
| 18–23 | 1,973 (4.0) |
| ≥24 | 3,381 (6.9) |
| Median (IQR) | 4 (2–9) |

BMI, body mass index; IPI, interpregnancy interval; IQR, interquartile range.

There was no evidence of increased risks of adverse pregnancy outcomes associated with an IPI >12 months (*p* > 0.05) except for GDM. For GDM, compared to IPI of 6 to 11 months (4.5%), there were increased risks for births after 12 to 17 months (5.8%) (aRR 1.20, 95% CI: 1.02, 1.40, *p* = 0.02), 18 to 23 months (aRR 1.24, 95% CI: 1.02, 1.50), but not for IPI ≥24

**Table 2. Characteristics for women with births after induced abortion at the time of birth after the IPI for births between 2008 and 2016 in Norway (*n* = 23,707).**

| Variable | Number (%) |
|---|---|
| **Maternal age (in years)** | |
| 14–19 | 734 (3.1) |
| 20–24 | 5,531 (23.3) |
| 25–29 | 8,041 (33.9) |
| 30–34 | 5,944 (25.1) |
| 35–39 | 2,782 (11.7) |
| ≥40 | 675 (2.9) |
| Median (IQR) | 28 (24–48) |
| **Gravidity** | |
| 1 | 15,082 (63.6) |
| 2 | 5,233 (22.0) |
| 3+ | 3,402 (14.4) |
| **Prepregnancy BMI (kg/m$^2$)** | |
| 18.5 | 765 (3.2) |
| 18.5–25 | 9,523 (40.2) |
| 25–30 | 3.277 (13.8) |
| ≥30 | 1,583 (6.9) |
| Missing | 8,559 (36.1) |
| **Smoking during pregnancy** | |
| No | 16,702 (70.5) |
| Yes | 4,017 (16.9) |
| Missing | 2,988 (12.6) |
| **Birth year** | |
| 2008 | 20 (0.1) |
| 2009 | 933 (3.9) |
| 2010 | 1,930 (8.1) |
| 2011 | 2,576 (10.9) |
| 2012 | 3,154 (13.3) |
| 2013 | 3,455 (14.6) |
| 2014 | 3,761 (15.9) |
| 2015 | 3,879 (16.4) |
| 2016 | 3,999 (16.9) |
| **IPI in months** | |
| <3 | 1,633 (6.9) |
| <6 | 4,574 (19.3) |
| 6–11 | 4,244 (17.9) |
| 12–17 | 3,255 (13.7) |
| 18–23 | 2,540 (10.7) |
| ≥24 | 9,094 (38.4) |
| Median (IQR) | 17 (7–34) |

BMI, body mass index; IPI, interpregnancy interval; IQR, interquartile range.

months (aRR 1.14, 95% CI: 0.97, 1.34, *p* = 0.10) (Table 3). The corresponding *e*-values were low, ranging between 1.11 and 1.83, for the associations between IPI and all adverse outcomes after previous miscarriage.

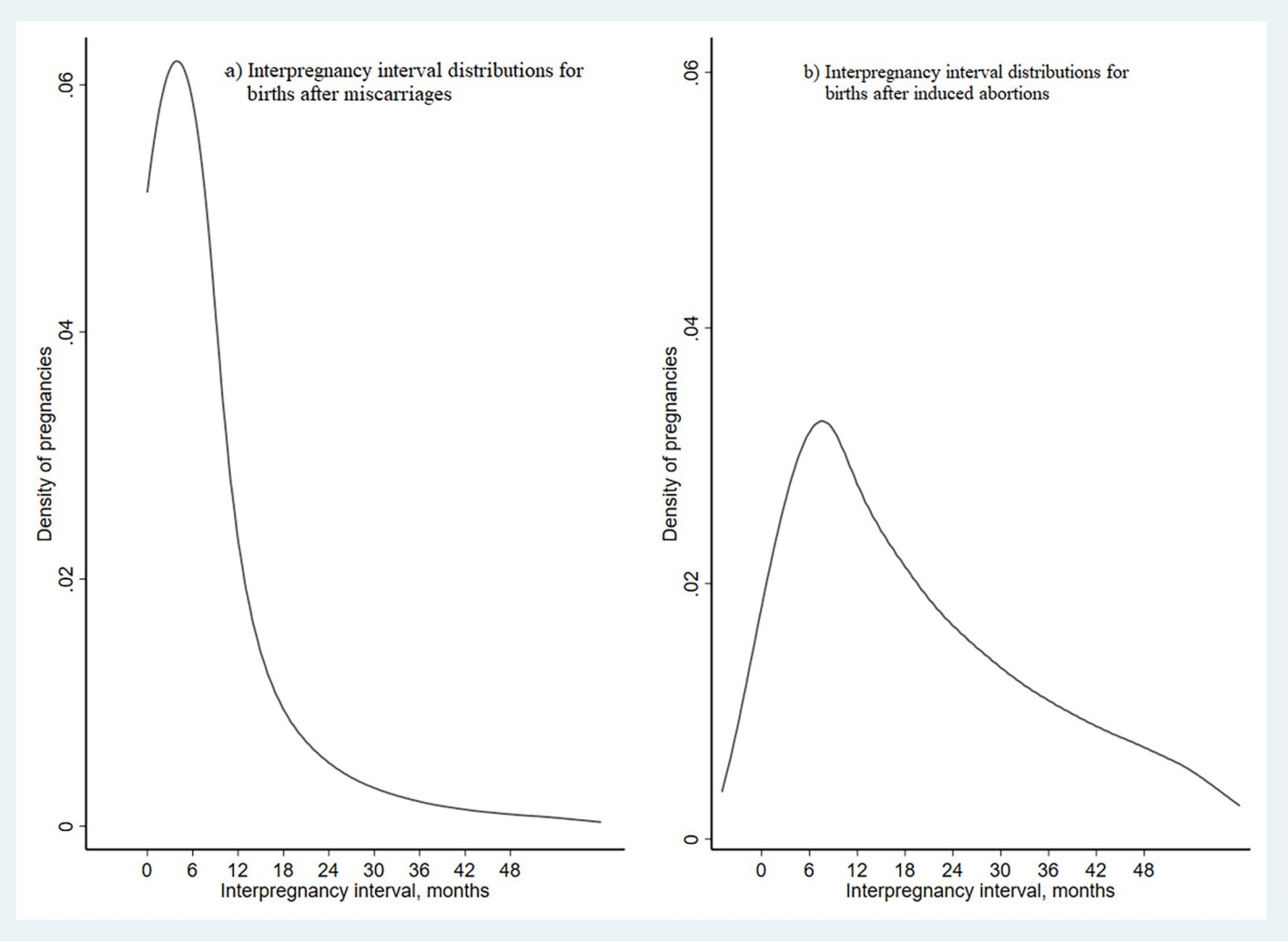

**Fig 2. Kernel density distributions of interpregnancy interval after miscarriages and induced abortions for births between 2008 and 2016 in Norway.**

### Previous induced abortion

The median IPI after a previous induced abortion was 17 months (IQR = 7 to 34 months), where 19% ($n$ = 4,574) of births were conceived within 6 months, and 18% ($n$ = 4,244) were conceived between 6 and 11 months (Table 2 and Fig 2). There were 5.5% PTB, 3.2% spontaneous PTB, 10.8% SGA births, 8.8% LGA births, 2.8% preeclampsia, and 3.2% GDM among births among births following induced abortions (S2 Table).

Compared with births following 6 to 11 months of IPI, there was a modest increased risk of SGA after an IPI of <3 months with a confidence interval spanning the null value (aRR 1.16, 95% CI: 0.99, 1.37, $p$ = 0.07), and a modest decreased risk of LGA among those with an IPI of 3 to 5 months (aRR 0.84, 95% CI: 0.72, 0.98, $p$ = 0.03). For IPI >12 months, there was no evidence of risk of adverse pregnancy outcomes (Table 4). Corresponding $e$-values were low, ranging between 1.21 and ≤2.55, for the associations between IPI and adverse outcomes after previous induced abortion.

**Table 3. IPI after previous miscarriage and risk of adverse pregnancy outcomes for births between 2008 and 2016 in Norway (n = 49,058).**

| Outcome | IPI | Number of cases (%) | RR (95% CI) | aRR (95% CI)* | p-value for aRR | E-value aRR** (lower 95% CI) |
|---|---|---|---|---|---|---|
| **PTB (n = 49,058)** | <3 m | 1,002 (5.8) | 0.94 (0.86, 1.04) | 0.96 (0.87, 1.06) | 0.40 | 1.25 (1) |
| | 3–5 m | 719 (5.6) | 0.91 (0.82, 1.01) | 0.92 (0.83, 1.02) | 0.12 | 1.39 (1) |
| | 6–11 m | 615 (6.2) | Ref | Ref | | Ref |
| | 12–17 m | 257 (7.0) | 1.14 (0.99, 1.31) | 1.14 (0.99, 1.31) | 0.08 | 1.54 (1) |
| | 18–23 m | 150 (7.6) | 1.23 (1.04, 1.45) | 1.23 (1.04, 1.46) | 0.02 | 1.76 (1.24) |
| | ≥24 m | 230 (6.8) | 1.10 (0.95, 1.28) | 1.14 (0.98, 1.32) | 0.09 | 1.54 (1) |
| **Spontaneous PTB (n = 47,780)** | <3 m | 592 (3.5) | 0.98 (0.86, 1.11) | 0.97 (0.85, 1.11) | 0.64 | 1.21 (1) |
| | 3–5 m | 403 (3.2) | 0.89 (0.78, 1.03) | 0.89 (0.78, 1.03) | 0.11 | 1.50 (1) |
| | 6–11 m | 350 (3.6) | Ref | Ref | | Ref |
| | 12–17 m | 137 (3.9) | 1.08 (0.89, 1.31) | 1.08 (0.89, 1.31) | 0.43 | 1.37 (1) |
| | 18–23 m | 86 (4.5) | 1.25 (0.99, 1.57) | 1.26 (1.00, 1.59) | 0.05 | 1.83 (1) |
| | ≥24 m | 127 (3.9) | 1.07 (0.88, 1.31) | 1.12 (0.92, 1.37) | 0.26 | 1.49 (1) |
| **SGA (n = 49,058)** | <3 m | 1,483 (8.6) | 0.85 (0.79, 0.92) | 0.85 (0.79, 0.92) | 0.00 | 1.63 (1.39) |
| | 3–5 m | 1,157 (9.0) | 0.89 (0.82, 0.97) | 0.89 (0.82, 0.96) | 0.01 | 1.50 (1.25) |
| | 6–11 m | 1,008 (10.1) | Ref | Ref | | Ref |
| | 12–17 m | 420 (11.5) | 1.14 (1.02, 1.27) | 1.13 (1.02, 1.26) | 0.03 | 1.51 (1.16) |
| | 18–23 m | 197 (10.0) | 0.99 (0.85, 1.14) | 0.97 (0.84, 1.12) | 0.70 | 1.21 (1) |
| | ≥24 m | 389 (11.5) | 1.14 (1.02, 1.27) | 1.09 (0.97, 1.21) | 0.15 | 1.40 (1) |
| **LGA (n = 49,058)** | <3 m | 1,809 (10.5) | 1.02 (0.95, 1.10) | 1.04 (0.97, 1.12) | 0.30 | 1.24 (1) |
| | 3–5 m | 1,312 (10.2) | 1.00 (0.93, 1.08) | 1.01 (0.93, 1.09) | 0.82 | 1.11 (1) |
| | 6–11 m | 1,021 (10.2) | Ref | Ref | | Ref |
| | 12–17 m | 390 (10.7) | 1.04 (0.93, 1.16) | 1.05 (0.94, 1.17) | 0.34 | 1.28 (1) |
| | 18–23 m | 197 (10.0) | 0.98 (0.84, 1.13) | 0.99 (0.86, 1.15) | 0.92 | 1.11 (1) |
| | ≥24 m | 314 (9.3) | 0.91 (0.80, 1.03) | 0.95 (0.85, 1.08) | 0.45 | 1.29 (1) |
| **Preeclampsia (n = 49,058)** | <3 m | 512 (3.0) | 0.91 (0.80, 1.05) | 0.93 (0.81, 1.07) | 0.29 | 1.36 (1) |
| | 3–5 m | 391 (3.1) | 0.94 (0.81, 1.08) | 0.95 (0.82, 1.09) | 0.46 | 1.29 (1) |
| | 6–11 m | 324 (3.4) | Ref | Ref | | Ref |
| | 12–17 m | 133 (3.7) | 1.12 (0.92, 1.37) | 1.11 (0.91, 1.35) | 0.32 | 1.46 (1) |
| | 18–23 m | 72 (3.6) | 1.12 (0.87, 1.44) | 1.11 (0.86, 1.42) | 0.43 | 1.46 (1) |
| | ≥24 m | 114 (3.4) | 1.04 (0.84, 1.28) | 1.01 (0.82, 1.25) | 0.92 | 1.11 (1) |
| **GDM (n = 49,058)** | <3 m | 571 (3.3) | 0.74 (0.66, 0.84) | 0.85 (0.75, 0.95) | 0.01 | 1.67 (1.29) |
| | 3–5 m | 508 (4.0) | 0.89 (0.79, 1.00) | 0.95 (0.84, 1.08) | 0.43 | 1.29 (1) |
| | 6–11 m | 446 (4.5) | Ref | Ref | | Ref |
| | 12–17 m | 211 (5.8) | 1.29 (1.10, 1.51) | 1.20 (1.02, 1.40) | 0.02 | 1.69 (1.16) |
| | 18–23 m | 122 (6.2) | 1.38 (1.14, 1.68) | 1.24 (1.02, 1.50) | 0.03 | 1.79 (1.16) |
| | ≥24 m | 216 (6.4) | 1.43 (1.22, 1.67) | 1.14 (0.97, 1.34) | 0.10 | 1.54 (1) |

aRR, adjusted relative risk; CI, confidence interval; GDM, gestational diabetes mellitus; IPI, interpregnancy interval; LGA, large for gestational age; PTB, preterm birth; RR, relative risk; SGA, small for gestational age.

*Adjusted for maternal age, gravidity, and year of birth at the time of birth after interval. For maternal age, we used restricted cubic splines with 5 knots placed at the 5th, 27.5th, 50th, 72.5th, and 95th percentiles in the study population, which corresponds to 21, 26, 30, 33, and 40 years.

**E-values for unmeasured confounding for the association between IPI after miscarriage and induced abortion and adverse pregnancy outcomes.

## Sensitivity analysis

Additional adjustment for smoking during pregnancy and prepregnancy BMI did not alter our conclusions (S3 and S4 Tables). Our results adjusting for covariates measured at the time of miscarriage or induced abortion (before interval) provided consistent findings (S5 and

**Table 4. IPI after previous induced abortion and risk of adverse pregnancy outcomes for births between 2008 and 2016 in Norway (n = 23,707).**

| Outcome | IPI | Number of cases (%) | RR (95% CI) | aRR (95% CI)* | p-value for aRR | E-value aRR** (lower 95% CI) |
|---|---|---|---|---|---|---|
| **PTB (n = 23,707)** | <3 m | 110 (6.7) | 1.20 (0.97, 1.49) | 1.17 (0.94, 1.46) | 0.15 | 1.62 (1) |
| | 3–5 m | 157 (5.3) | 0.95 (0.78, 1.16) | 0.94 (0.77, 1.14) | 0.53 | 1.32 (1) |
| | 6–11 m | 238 (5.6) | Ref | Ref | | Ref |
| | 12–17 m | 150 (4.6) | 0.82 (0.67, 1.00) | 0.84 (0.69, 1.02) | 0.08 | 1.67 (1) |
| | 18–23 m | 134 (5.3) | 0.94 (0.77, 1.16) | 0.97 (0.79, 1.20) | 0.80 | 1.21 (1) |
| | ≥24 m | 506 (5.6) | 1.00 (0.85, 1.15) | 1.10 (0.94, 1.30) | 0.22 | 1.43 (1) |
| **Spontaneous PTB (n = 23,163)** | <3 m | 62 (3.9) | 1.15 (0.86, 1.54) | 1.13 (0.85, 1.52) | 0.41 | 1.51 (1) |
| | 3–5 m | 102 (3.5) | 1.04 (0.81, 1.34) | 1.03 (0.80, 1.32) | 0.83 | 1.21 (1) |
| | 6–11 m | 141 (3.4) | Ref | Ref | | Ref |
| | 12–17 m | 67 (2.1) | 0.62 (0.47, 0.83) | 0.63 (0.47, 0.84) | 0.00 | 2.55 (1.56) |
| | 18–23 m | 84 (3.4) | 0.99 (0.76, 1.29) | 1.02 (0.78, 1.33) | 0.90 | 1.16 (1) |
| | ≥24 m | 295 (3.3) | 0.98 (0.80, 1.19) | 1.06 (0.86, 1.31) | 0.56 | 1.31 (1) |
| **SGA (n = 23,707)** | <3 m | 188 (11.5) | 1.16 (0.98, 1.36) | 1.16 (0.99, 1.37) | 0.07 | 1.59 (1) |
| | 3–5 m | 318 (10.8) | 1.08 (0.95, 1.24) | 1.09 (0.95, 1.25) | 0.23 | 1.40 (1) |
| | 6–11 m | 423 (10.0) | Ref | Ref | | Ref |
| | 12–17 m | 363 (11.2) | 1.12 (0.98, 1.28) | 1.12 (0.98, 1.28) | 0.09 | 1.49 (1) |
| | 18–23 m | 269 (10.6) | 1.06 (0.92, 1.23) | 1.07 (0.93, 1.24) | 0.37 | 1.34 (1) |
| | ≥24 m | 997 (11.0) | 1.10 (0.99, 1.23) | 1.12 (1.00, 1.26) | 0.05 | 1.49 (1) |
| **LGA (n = 23,707)** | <3 m | 152 (9.3) | 0.99 (0.83, 1.18) | 0.98 (0.82, 1.17) | 0.80 | 1.16 (1) |
| | 3–5 m | 234 (8.0) | 0.84 (0.72, 0.99) | 0.84 (0.72, 0.98) | 0.03 | 1.67 (1.16) |
| | 6–11 m | 400 (9.4) | Ref | Ref | | Ref |
| | 12–17 m | 307 (9.4) | 1.00 (0.87, 1.15) | 1.00 (0.87, 1.15) | 0.99 | 1.00 (1) |
| | 18–23 m | 205 (8.1) | 0.86 (0.73, 1.01) | 0.85 (0.73, 1.01) | 0.06 | 1.63 (1) |
| | ≥24 m | 811 (8.9) | 0.95 (0.84, 1.06) | 0.95 (0.84, 1.07) | 0.41 | 1.29 (1) |
| **Preeclampsia (n = 23,707)** | <3 m | 50 (3.1) | 1.13 (0.81, 1.57) | 1.15 (0.83, 1.60) | 0.40 | 1.57 (1) |
| | 3–5 m | 80 (2.7) | 1.00 (0.76, 1.33) | 1.02 (0.77, 1.35) | 0.90 | 1.16 (1) |
| | 6–11 m | 115 (2.7) | Ref | Ref | | Ref |
| | 12–17 m | 80 (2.5) | 0.91 (0.68, 1.20) | 0.90 (0.68, 1.20) | 0.49 | 1.46 (1) |
| | 18–23 m | 63 (2.5) | 0.92 (0.69, 1.24) | 0.91 (0.67, 1.23) | 0.53 | 1.43 (1) |
| | ≥24 m | 277 (3.1) | 1.12 (0.91, 1.39) | 1.09 (0.87, 1.36) | 0.46 | 1.40 (1) |
| **GDM (n = 23,707)** | <3 m | 36 (2.2) | 0.75 (0.52, 1.08) | 0.75 (0.52, 1.08) | 0.13 | 2.00 (1) |
| | 3–5 m | 76 (2.6) | 0.88 (0.66, 1.16) | 0.88 (0.67, 1.16) | 0.37 | 1.53 (1) |
| | 6–11 m | 125 (3.0) | Ref | Ref | | Ref |
| | 12–17 m | 98 (3.0) | 1.02 (0.79, 1.33) | 0.97 (0.75, 1.25) | 0.81 | 1.21 (1) |
| | 18–23 m | 63 (2.5) | 0.84 (0.62, 1.14) | 0.79 (0.59, 1.06) | 0.12 | 1.85 (1) |
| | ≥24 m | 354 (3.9) | 1.32 (1.08, 1.62) | 1.04 (0.85, 1.28) | 0.72 | 1.24 (1) |

aRR, adjusted relative risk; CI, confidence interval; IPI, interpregnancy interval; GDM, gestational diabetes mellitus; LGA, large for gestational age; PTB, preterm birth; RR, relative risk; SGA, small for gestational age.

*Adjusted for maternal age, gravidity, and year of birth at the time of birth after interval. For maternal age, we used restricted cubic splines with 5 knots placed at the 5th, 27.5th, 50th, 72.5th, and 95th percentiles in the study population, which corresponds to 20, 25, 28, 32, and 38 years.

**E-values for unmeasured confounding for the association between IPI after miscarriage and induced abortion and adverse pregnancy outcomes.

S6 Tables). Our analysis combining <3 months and 3 to 5 months of IPI into a <6 months category showed consistent results with estimates for IPI <3 months category (S7 and S8 Tables). Our results restring births from women with only 1 miscarriage or induced abortion in the cohort did not change our results (S9 and S10 Tables)

## Discussion

We used a Norwegian national registry-linkage to investigate the association between IPI following a miscarriage or induced abortion with adverse pregnancy outcomes in the subsequent birth. Contrary to the current WHO recommendations advising women to wait a minimum of 6 months after miscarriages or induced abortions [9], we found no evidence of elevated risk of PTB, spontaneous PTB, LGA, or preeclampsia among women with a very short (<3) or short (<6 months) IPIs after a miscarriages or induced abortions. Rather, we found that the risk of SGA was lower among women with births following an IPI of <3 months and 3 to 5 months after miscarriage. We also found a lower risk of GDM among women with very short (<3 months) IPI after miscarriage. There was no evidence of higher risks of adverse pregnancy outcomes for IPIs >12 months after miscarriages or induced abortions except for a higher risk of GDM for births after miscarriages with IPIs between 12 and 23 months.

Our study is consistent with previous studies from Scotland [10] and the United States [15–17], which reported no difference or reduced risk of adverse pregnancy outcomes following short (<6 months) IPI after a previous miscarriage. Although not directly comparable to our study due to only including births after first trimester miscarriages and having a very small sample size (*n* = 107), our results were consistent with a study from Israel that indicated that conception shortly after a spontaneous miscarriage was not associated with adverse maternal or neonatal outcomes [31]. Our results differ from a large cohort study conducted in Latin America, which did not distinguish between miscarriages and induced abortions reporting an elevated risk of adverse pregnancy outcomes among women with an IPI of <6 months [8]. On the other hand, another study from the US considering births after miscarriages and induced abortions together showed a reduced risk of PTB [11]. Previous studies from Finland [12] and China [32] investigating the effect of IPI after induced abortions showed increased risks of PTB and SGA after short (<6 months) IPI. Unlike these studies, we observed no evidence of elevated risks of either PTB or SGA according to IPI following an induced abortion. Apart from one study from Finland [12], our study was not in agreement with previous studies that reported elevated risks of adverse pregnancy outcomes for those pregnancies with long (>24 months) IPI after miscarriages as compared to IPI 6 to 11 months [10,33,34]. As our study included women with at least 2 pregnancies occurring in less than a decade, we might not have observed adequate number of births with longer IPIs to observe any associations. There were also some differences between our study and previous studies where some studies used 18 to 23 months as reference [8,11,32], included smaller sample sizes [12,15,17,18,32], and relied on self-reported data [8,32].

Although the proportion of short (<6 months) IPI were generally high following both miscarriages and induced abortions, the proportion of short IPI that followed miscarriages (61%) were much higher than the proportion of short IPI that followed induced abortions (19%). A previous study reported over one-third of births (37%) following a stillbirth were conceived within the first 6 months [35]. However, for IPIs after live births, a study that involved 4 high-income countries including Norway showed that less than 5% of conceptions following live births occurred in the first 6 months [36]. Our findings, in combination with the previous findings of Regan and colleagues [35] indicate that women with early and late pregnancy losses (miscarriages or stillbirths) attempt pregnancy more quickly than those women whose pregnancies ended with live births. On the other hand, IPIs after induced abortions could be relatively longer if previous terminations resulted from unintended pregnancy, and, hence, these women could adopt contraceptive methods as part of routine postabortal care [37].

Our findings suggested that IPI <6 months following miscarriages may not increase the risks of adverse outcomes in the next pregnancy. One possible explanation would be that

pregnancies conceived shortly after a miscarriage are more likely to be intended, and, hence, these women may seek health services with the aim of avoiding the previous unfavourable experience [10]. The reduced risk of SGA after very short (<3 months) and short (3 to 5 months) IPI in our study may support the previous hypothesis suggesting women who were able to conceive quickly after miscarriages might have better fecundity reflecting women's high reproductive fitness in comparison to women with longer IPI [16]. Despite the assumption that women with previous pregnancies resulting in a live birth could have had nutritional depletions, which are mostly occurring in the second trimester and early postpartum period through breastfeeding [38], it is plausible that women with miscarriages would not reach at a point when nutritional depletion starts as most miscarriages usually occur in the first 12 weeks [39,40].

A large study conducted in Latin America indicated an elevated risk of adverse pregnancy outcomes despite not separating miscarriage from induced abortion [8]. The authors hypothesized that their results could be attributed to reproductive tract infections due to the possibility of unsafe abortions, which may lead to negative consequences to the growing fetus in the subsequent pregnancy. However, our results indicated no elevated risk of adverse pregnancy outcomes in those pregnancies that followed induced abortions. This could be due to the provision of safe abortion services free of cost in the public health system in Norway, and, hence, the risks of infections are minimal [37]. Studies also indicated that uterine infections are mostly associated with surgical abortions than medical terminations. In Norway, medical abortions comprising more than 80% of all abortions conducted in 2013, and 95% of these terminations were performed in the first trimester [41]. Since 2009, medical abortion services were further expanded for women to get access to self-administered misoprostol to use at home for pregnancies terminations up to 12 weeks of gestation [42].

Previous studies did not assess the association between previous miscarriage and risk of GDM in the subsequent pregnancy according to interpregnancy interval. Two Chinese studies [43,44] reported that women who had previous miscarriage had a higher risk for developing gestational diabetes during subsequent pregnancies, although their studies did not evaluate IPI specifically. In our study, while we observed an increased risk of risk of GDM for births between 12 and 23 months of IPI, we found a reduced risk of GDM for births after a very short (<3 months) IPI. The underlying mechanisms for the associations are unclear and warrant further investigations. Yang and colleagues [44] speculated that the association between previous miscarriages and GDM might have occurred due to common risk factors or shared pathological mechanisms.

The strengths of our study are its ability to provide comprehensive information of early miscarriages and induced abortions from the combination of national health registries in Norway, in addition to be able to distinguish the risks of adverse pregnancy outcomes according to IPI after a miscarriage and an induced abortion, which was not possible in most settings [8,11]. Unlike most previous studies [10,26,34] that investigated IPIs of <6 months after miscarriage or induced abortions, but not all [15,16], our study investigated the effect of IPI for very short (<3 months) and short (3 to 5 months) IPIs. To allow comparison with literature and WHO recommendation, we also estimated risk of adverse pregnancy outcomes for IPI <6 months, although the results broadly concurred with the estimates for IPI <3 months. Although the gestational age at previous miscarriage or induced abortion affects the IPI, we did not have information on exact gestational age for those miscarriages or induced abortions occurred before 12 weeks, which comprised 99% of miscarriages and 96% of induced abortions. However, we did have information on gestational age at miscarriages or induced abortions after 12 weeks, and, hence, the risk of misclassification of exposure is minimal for these pregnancies. With regard to the timing of confounder measurement, it has recently been

recommended to adjust confounders for measures prior to IPI (i.e., at the time of the miscarriage or induced abortion), as these are the characteristics that may be associated both with IPI and adverse pregnancy outcomes [30,45]. Our sensitivity analysis adjusting for maternal age, gravidity, and year at the time of the miscarriage or induced abortion indicated no significant difference in the risk estimates from the main results. However, since smoking and prepregnancy BMI information were only available for pregnancies registered in the birth registry, we were not able to adjust for these variables at the time of miscarriage or induced abortion. These results were not unexpected because most of the pregnancies occurred within 1 year of miscarriage or induced abortion, and, hence, maternal characteristics including maternal age and socioeconomic status are unlikely to change significantly. Moreover, inherent to retrospective registry-based studies, we did not have information on potential confounders variables. For example, while pregnancy intention and health-seeking behaviour are likely to vary between women with longer pregnancies [46–48] but were unfortunately not available. Furthermore, we only had information on miscarriages that resulted in contact with the healthcare system. Since our study employed data from a single high-income country with better healthcare services, our results could not be generalisable to other settings with different population.

In conclusion, our study found that that conceiving within 6 months after a miscarriage or an induced abortion was not associated with increased risks of adverse pregnancy outcomes. In combination with previous research, these results suggest that women could attempt pregnancy soon after a previous miscarriage or induced abortion without increasing perinatal health risks. Our results do not support current international recommendations to wait at least 6 months after miscarriages or induced abortions.

## Supporting information

**S1 STROBE Checklist. STROBE checklist.**
(DOCX)

**S1 Table. Adverse pregnancy outcomes after miscarriages between 2008 and 2016 in Norway (*n* = 49,058).** GDM, gestational diabetes mellitus; LGA, large for gestational age; PTB, preterm birth; SGA, small for gestational age. *Births with nonspontaneous preterm outcomes were excluded when defining spontaneous PTB.
(DOCX)

**S2 Table. Adverse pregnancy outcomes after induced abortion between 2008 and 2016 in Norway (*n* = 23,707).** GDM, gestational diabetes mellitus; LGA, large for gestational age; PTB, preterm birth; SGA, small for gestational age. *Births with nonspontaneous preterm outcomes were excluded when defining spontaneous PTB.
(DOCX)

**S3 Table. Sensitivity analysis—IPI after previous miscarriage and risk of adverse pregnancy outcomes with complete information on maternal smoking during pregnancy and prepregnancy BMI for births between 2008 and 2016 in Norway (*n* = 27,747).** aRR, adjusted relative risk; BMI, body mass index; CI, confidence interval; GDM, gestational diabetes mellitus; IPI, interpregnancy interval; LGA, large for gestational age; PTB, preterm birth; RR, relative risk; SGA, small for gestational age. *Births with nonspontaneous preterm outcomes were excluded when defining spontaneous PTB. **Adjusted for maternal age, gravidity, year of birth, maternal smoking during pregnancy, and prepregnancy BMI at the time of birth after interval. For maternal age and prepregnancy BMI variables, we used restricted cubic splines with 5 knots placed at the 5th, 27.5th, 50th, 72.5th, and 95th percentiles in the study population, which corresponds to 21, 26, 30, 33, and 40 years for maternal age, and 19, 21, 24, 27, and

34 kg/m$^2$ for prepregnancy BMI.
(DOCX)

**S4 Table. Sensitivity analysis—IPI after previous induced abortion and risk of adverse pregnancy outcomes with complete information on maternal smoking during pregnancy and prepregnancy BMI for births between 2008 and 2016 in Norway ($n$ = 13,932).** aRR, adjusted relative risk; BMI, body mass index; CI, confidence interval; GDM, gestational diabetes mellitus; IPI, interpregnancy interval; LGA, large for gestational age; PTB, preterm birth; RR, relative risk; SGA, small for gestational age. *Births with nonspontaneous preterm outcomes were excluded when defining spontaneous PTB. **Adjusted for maternal age, parity, year of birth, maternal smoking during pregnancy, and prepregnancy BMI at the time of birth after interval. For maternal age and prepregnancy BMI variables, we used restricted cubic splines with 5 knots placed at the 5th, 27.5th, 50th, 72.5th, and 95th percentiles in the study population, which corresponds to 20, 25, 28, 32, and 38 years for maternal age, and 18, 21, 23, 26, and 33 kg/m$^2$ for prepregnancy BMI.
(DOCX)

**S5 Table. Sensitivity analysis—aRR for the association between IPI after a miscarriage and adverse pregnancy outcomes adjusted for covariates prior to IPI ($n$ = 49,058).** aRR, adjusted relative risk; BMI, body mass index; CI, confidence interval; GDM, gestational diabetes mellitus; IPI, interpregnancy interval; LGA, large for gestational age; PTB, preterm birth; RR, relative risk; SGA, small for gestational age. *Births with nonspontaneous preterm outcomes were excluded when defining spontaneous PTB. **Adjusted for maternal age, gravidity, and year of birth at the time of miscarriage (before interval). For maternal age, we used restricted cubic splines with 5 knots placed at the 5th, 27.5th, 50th, 72.5th, and 95th percentiles in the study population, which corresponds to 21, 26, 30, 33, and 40 years for births after a miscarriage.
(DOCX)

**S6 Table. Sensitivity analysis—aRR for the association between IPI after an induced abortion and adverse pregnancy outcomes adjusted for covariates prior to IPI ($n$ = 23,707).** aRR, adjusted relative risk. BMI, body mass index; CI, confidence interval; GDM, gestational diabetes mellitus; IPI, interpregnancy interval; LGA, large for gestational age; PTB, preterm birth; RR, relative risk; SGA, small for gestational age. *Births with nonspontaneous preterm outcomes were excluded when defining spontaneous PTB. **Adjusted for maternal age, gravidity, and year of birth at the time of miscarriage (before interval). For maternal age, we used restricted cubic splines with 5 knots placed at the 5th, 27.5th, 50th, 72.5th, and 95th percentiles in the study population, which corresponds to 18, 22, 25, 29, and 36 for births after an induced abortion.
(DOCX)

**S7 Table. Sensitivity analysis—IPI after previous miscarriage and risk of adverse pregnancy outcomes accounting <6 months of IPI category ($n$ = 49,058). aRR, adjusted relative risk.** BMI, body mass index; CI, confidence interval; GDM, gestational diabetes mellitus; IPI, interpregnancy interval; LGA, large for gestational age; PTB, preterm birth; RR, relative risk; SGA, small for gestational age. *Births with nonspontaneous preterm outcomes were excluded when defining spontaneous PTB. **Adjusted for maternal age, gravidity, and year of birth at the time of birth after interval. For maternal age, we used restricted cubic splines with 5 knots placed at the 5th, 27.5th, 50th, 72.5th, and 95th percentiles in the study population, which corresponds to 21, 26, 30, 33, and 40 years.
(DOCX)

**S8 Table. Sensitivity analysis—IPI after previous induced abortion and risk of adverse pregnancy outcomes accounting <6 months of IPI category (*n* = 23,707).** aRR, adjusted relative risk; BMI, body mass index; CI, confidence interval; GDM, gestational diabetes mellitus; IPI, interpregnancy interval; LGA, large for gestational age; PTB, preterm birth; RR, relative risk; SGA, small for gestational age. *Births with nonspontaneous preterm outcomes were excluded when defining spontaneous PTB. **Adjusted for maternal age, gravidity, and year of birth at the time of birth after interval. For maternal age, we used restricted cubic splines with 5 knots placed at the 5th, 27.5th, 50th, 72.5th, and 95th percentiles in the study population, which corresponds to 20, 25, 28, 32, and 38 years.
(DOCX)

**S9 Table. Sensitivity analysis—IPI after previous miscarriages and risk of adverse pregnancy outcomes among births from women with only 1 miscarriage in the cohort (*n* = 47,411).** aRR, adjusted relative risk; BMI, body mass index; CI, confidence interval; GDM, gestational diabetes mellitus; IPI, interpregnancy interval; LGA, large for gestational age; PTB, preterm birth; RR, relative risk; SGA, small for gestational age. *Births with nonspontaneous preterm outcomes were excluded when defining spontaneous PTB. *Adjusted for maternal age, gravidity, and year of birth at the time of birth after interval. For maternal age, we used restricted cubic splines with 5 knots placed at the 5th, 27.5th, 50th, 72.5th, and 95th percentiles in the study population, which corresponds to 21, 26, 30, 33, and 40 years. **E-values for unmeasured confounding for the association between IPI after miscarriage and induced abortion and adverse pregnancy outcomes.
(DOCX)

**S10 Table. Sensitivity analysis—IPI after previous induced abortion and risk of adverse pregnancy outcomes among births from women with only 1 induced abortion in the cohort (*n* = 23,185).** aRR, adjusted relative risk; BMI, body mass index; CI, confidence interval; GDM, gestational diabetes mellitus; IPI, interpregnancy interval; LGA, large for gestational age; PTB, preterm birth; RR, relative risk; SGA, small for gestational age. *Births with nonspontaneous preterm outcomes were excluded when defining spontaneous PTB. **Adjusted for maternal age, gravidity, and year of birth at the time of birth after interval. For maternal age, we used restricted cubic splines with 5 knots placed at the 5th, 27.5th, 50th, 72.5th, and 95th percentiles in the study population, which corresponds to 20, 25, 28, 32, and 38 years. **E-values for unmeasured confounding for the association between IPI after miscarriage and induced abortion and adverse pregnancy outcomes.
(DOCX)

## Author Contributions

**Conceptualization:** Gizachew A. Tessema, Siri E. Håberg, Gavin Pereira, Maria C. Magnus.

**Data curation:** Gizachew A. Tessema, Maria C. Magnus.

**Formal analysis:** Gizachew A. Tessema.

**Funding acquisition:** Gizachew A. Tessema, Siri E. Håberg, Gavin Pereira.

**Investigation:** Gizachew A. Tessema, Annette K. Regan, Jennifer Dunne, Maria C. Magnus.

**Methodology:** Gizachew A. Tessema, Siri E. Håberg, Gavin Pereira, Annette K. Regan, Jennifer Dunne, Maria C. Magnus.

**Project administration:** Gizachew A. Tessema.

**Software:** Gizachew A. Tessema, Siri E. Håberg.

**Supervision:** Siri E. Håberg, Gavin Pereira, Annette K. Regan, Maria C. Magnus.

**Validation:** Gizachew A. Tessema, Maria C. Magnus.

**Visualization:** Gizachew A. Tessema, Maria C. Magnus.

**Writing – original draft:** Gizachew A. Tessema.

**Writing – review & editing:** Gizachew A. Tessema, Siri E. Håberg, Gavin Pereira, Annette K. Regan, Jennifer Dunne, Maria C. Magnus.

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
