## [Editor Report · Decision Letter 0]

6 May 2022

Dear Dr Tessema, 

Thank you for submitting your manuscript entitled "Interpregnancy interval and adverse pregnancy outcomes among pregnancies following miscarriages or induced abortions" for consideration by PLOS Medicine.

Your manuscript has now been evaluated by the PLOS Medicine editorial staff and I am writing to let you know that we would like to send your submission out for external peer review.

Please re-submit your manuscript within two working days, i.e. by May 10 2022 11:59PM.

Kind regards,

Louise Gaynor-Brook, MBBS PhD

Senior Editor

PLOS Medicine

---

## [Decision Letter · Decision Letter 1]

17 Aug 2022

Dear Dr. Tessema,

Thank you very much for submitting your manuscript "Interpregnancy interval and adverse pregnancy outcomes among pregnancies following miscarriages or induced abortions" (PMEDICINE-D-22-01509R1) for consideration at PLOS Medicine. 

[LINK]

In light of these reviews, I am afraid that we will not be able to accept the manuscript for publication in the journal in its current form, but we would like to consider a revised version that addresses the reviewers' and editors' comments. Obviously we cannot make any decision about publication until we have seen the revised manuscript and your response, and we plan to seek re-review by one or more of the reviewers. 

We expect to receive your revised manuscript by Sep 07 2022 11:59PM. Please email us (plosmedicine@plos.org) if you have any questions or concerns.

We look forward to receiving your revised manuscript. 

Sincerely,

Philippa Dodd, MBBS MRCP PhD

PLOS Medicine

pdodd@plos.org

plosmedicine.org

GENERAL

Please revise your title to include the study design in the subtitle such as: “Interpregnancy interval and adverse pregnancy outcomes among pregnancies following miscarriages and induced abortions: a cohort study” 

As the data are not freely available, please describe briefly the ethical, legal, or contractual restriction that prevents you from sharing it. Please also include an appropriate contact (web or email address) for inquiries (this cannot be a study author).

Thank you for including an author summary we would suggest emphasizing as a final point in “what do these findings mean” that the results may prompt a review of guidelines, as we understand it

There are a number of typographical and grammatical errors throughout as detailed below and highlighted by the reviewers. Please check through the manuscript carefully for these errors

ABSTRACT METHODS and FINDINGS

Please include the main outcome measures and define any abbreviations used

Please quantify the main results with p values as well as 95% CIs

Please include the absolute risk(s) of relevant outcomes, not just relative risks (an example for absolute risk inclusion in the abstract can be found here: PMID: 28399126). 

Please include the important dependent variables that are adjusted for in the analyses.

In the last sentence of the Abstract Methods and Findings section, please describe the main limitation(s) of the study's methodology.

MAIN MANUSCRIPT METHODS AND RESULTS

Participants and exclusions – line 3 please check grammar … “we left with” suggest adding “were”

Please provide p values as well as 95% CIs, where a p value is given please also specify the statistical test used to determine it

TABLES

Please define the numbers in brackets in table 1 [i.e. 384 (0.8)] ?median

Please define the numbers in brackets in table 2 [i.e 47,308 (94.0) ? percentage

Please provide the unadjusted comparisons as well as the adjusted comparisons in Table S3

FIGURES

Please provide titles and legends for each individual table and figures in the Supporting Information.

DISCUSSION 

Some sentences throughout might benefit from shortening and re-wording to improve clarity and readability for example, in the Strengths and limitations of the study – the sentence on page 19 “…information were only available for pregnancies ending in the birth registry…” is a little confusing 

Comments from the reviewers:

Reviewer #1: I confine my remarks to statistical aspects of this paper. The general approach is fine, and I found some aspects commendable, but I do have some issues to resolve before I can recommend publication

NOTE: Line numbers would have made the review process easier

p. 2 "Log binomial" isn't the usual term. I would use "logistic", but this is not a big deal.

Not a stat issue, but SGA and other acronyms should be spelled out at first use. 

p. 5 Where possible, please give the effect size estimates from the various studies you report on. The fact that some studies found a significant effect and others did not is not enough. See Andrew Gelman's paper "The difference between significant and nonsignificant is not, itself, significant". Also, where you say "no increased risk" please insert "significant" (or, if they found a decrease, say that).

p. 7 Do not categorize IPI. Doing so increases type I and type II error and introduces a kind of magical thinking. that something amazing happens at the cutpoints. See my article https://medium.com/@peterflom/what-happens-when-we-categorize-an-independent-variable-in-regression-77d4c5862b6c (the article is about linear regression, but applies to logistic as well). If you wish to report results by category, for comparison purposes, that's OK, but the analysis should use IPI as a number, and maybe use splines to look at nonlinearity. Splines could even find that the categories are bad ones.

p. 8 Same issue for age and BMI as for IPI. BMI could easily have nonlinear effects. 

 Congratulations for doing some sensitivity analyses. Not enough studies include them

 I also commend the authors for not using stepwise or other automatic variable selection methods. They are deeply flawed.

How were the reference categories for the categorized continuous variables selected? I don't like categorization (see above) but, if you do categorize, there should be some stated reasons for choosing a reference category. The most usual choices are either the most common category or an extreme category. For IPI, both of those would suggest using < 3 month. Also, table 3 shows another problem with categorization - some of the cells are small sized, and the sizes vary a lot, so, the same OR can be significant in one category and not another.

Finally, figure 2 seems to be based on using IPI as a continuous measure. Both of these show very nonlinear relationships. This makes use of splines for IPI more important.

Reviewer #3: Delighted to review this manuscript that adds to the literature supporting early conception after miscarriage or induced abortion and calls into question WHO's guidance published in 2007.

Abstract: Please mention that only births following previous miscarriage/ induced abortion were looked at and not early pregnancy outcomes.

Author summary: "Available small studies exploring pregnancy outcomes after miscarriage (n<1,100) focused on the limited pregnancy outcomes such as recurrent miscarriage and preterm birth"- is this true? Some of the references cited have used larger datasets and the systematic review included many more pregnancies! This needs to be amended in the introduction section also. I feel the biggest strength of this study is in being able to look at IPI of <3 months and study outcomes following both miscarriage and induced abortion in the same population. "Our study suggests that conceiving within 6 months after a miscarriage or an induced abortion is not associated with increased risks of adverse pregnancy outcomes" - this should be reported in "what did the researchers find?" Perhaps highlight that this was true for IPI less than 3 months too?

Methods: The ICD codes used to identify miscarriage include Hydatidiform mole - these have completely different management and reproductive trajectory and should not be included in miscarriage. In the big scheme of things the numbers are probably too few to make any difference but if possible, the analyses should be repeated excluding these cases. Similarly, threatened abortion does not necessarily end in a miscarriage - in fact the vast majority of them don't. Not clear how more than one birth could have contributed to the index birth? Statistical analysis is appropriate and described well but could perhaps benefit from justifying why log binomial regression was used and relative risks calculated as effect measure particularly since the covariates adjusted for were at the time of the index pregnancy.

Results: I would prefer to see tables 1 and 2 separately for miscarriage and induced abortion with the characteristics and outcomes presented by IPI. As such the reader gets no idea as to what number (%) of outcomes occurred in each IPI category. Although, these are presented in table 3 and so table 2 is perhaps redundant? Table 3 should mention which covariates were adjusted for to calculate ARR. were the same covariates adjusted for in all the models? What was the rationale behind including these?

Discussion: A main strength of this study is the ability to study separately the effects of increasing IPI after miscarriage and induced abortion in the same population. This inevitably begs the question as to how confident researchers were that there was no or minimal misclassification. Looking at the codes used to identify the cases i am not convinced that there was not some mixing. However, given that similar findings were observed in both groups, this probably does not matter. Another strength is the ability to study outcomes in IPI <3 months. This again begs the question as to how confident the researchers were that the two pregnancies were not the same, particularly as i note that threatened miscarriage and bleeding in pregnancy were included as miscarriage. A recent publication had suggested that even the current guidance of waiting for at least one normal period after a miscarriage was not justified. Would the authors like to comment on that vis a vis guidance in Norway? Lee Reicher, Ronni Gamzu, Yuval Fouks, Ofer Isakov, Yariv Yogev, Sharon Maslovitz,The effects of a postmiscarriage menstrual period prior to reconceiving, American Journal of Obstetrics and Gynecology, Volume 223, Issue 3,2020, Pages 444.e1-444.e5,ISSN 0002-9378,

https://doi.org/10.1016/j.ajog.2020.06.051.

I would urge the authors to proof read the manuscript carefully as there are several missed words and typographical errors throughout. Some references are also not formatted correctly (Se ref 11)

Reviewer #4: Thank you for this well-written paper. I see this is a revision but cannot seem to access previous reviewer comments or responses. I have no further comments.

[LINK]

---

## [Decision Letter · Decision Letter 2]

14 Oct 2022

Dear Dr. Tessema,

Thank you very much for re-submitting your manuscript "Interpregnancy interval and adverse pregnancy outcomes among pregnancies following miscarriages or induced abortions in Norway (2008-2016): a cohort study" (PMEDICINE-D-22-01509R2) for review by PLOS Medicine.

I have discussed the paper with my colleagues and the academic editor and it was also seen again by 2 reviewers. I am pleased to say that provided the remaining editorial and production issues are dealt with we are planning to accept the paper for publication in the journal.

[LINK]

We look forward to receiving the revised manuscript by Oct 21 2022 11:59PM.   

Sincerely,

Philippa Dodd, MBBS MRCP PhD

PLOS Medicine

plosmedicine.org

Requests from Editors:

Thank you for the opportunity to handle your manuscript and for your careful and considerate attention in addressing the previous reviewer and editor comments. There are some further minor revisions detailed below that require your attention prior to publication.

GENERAL

Thank you for reporting according to STROBE. Please add the following statement, or similar, to the Methods: "This study is reported as per the Strengthening the Reporting of Observational Studies in Epidemiology (STROBE) guideline (S1 Checklist)." 

ABSTRACT

Thank you for revising the abstract. 

Thank you for including p-values. When reporting p-values please replace “p-value =…” with p = 0.07”. Please report p as <0.01 as opposed to p=0.00. Please check and amend throughout the abstract and main manuscript text

Thank you for including limitations of your data set. Please also include limitations of the study methodology (i.e. retrospective, observational etc), any problems with the use of registry data (reliability)? 

AUTHOR SUMMARY

Thank you for revising the author summary

I suggest “review of current international….” instead of “revision”

METHODS and RESULTS

Table 1: Thank you making revisions. I suggest a few further thing. There seems to be a redundant third unfilled column on the right please remove this. Suggest relabeling the heading to column 2: “Number of participants” or something similar. Perhaps also include the total number of ppts in the table in an appropriate place. 

DISCUSSION

Please remove sub headings and include the strengths and limitations section within the discussion which should be structured as follows: a short, clear summary of the article's findings; what the study adds to existing research and where and why the results may differ from previous research; strengths and limitations of the study; implications and next steps for research, clinical practice, and/or public policy; one-paragraph conclusion.

Please remove the funding statement and COI from the end of the manuscript and include only in the submission form

REFERENCES

Please see our website for reference guidelines https://journals.plos.org/plosmedicine/s/submission-guidelines#loc-references

Please ensure journal name abbreviations are those found in the National Center for Biotechnology Information (NCBI) databases. 

To help us extend the reach of your research, please provide any Twitter handle(s) that would be appropriate to tag, including your own, your coauthors’, your institution, funder, or lab. Please respond to this email with any handles you wish to be included when we tweet this paper.

Comments from the academic editor:

I have one remaining thing that I think they need to clarify. Their response to comment 4.2 in relation to gestational age (page 11) is, for me, inadequate. They refer to the gestational age of the miscarriage or induced abortion as a "confounder". But it is not. The gestational age at the time of the miscarriage or induced abortion is a determinant of the exposure. These events are broadly classified into first trimester and second trimester. The vast majority of both are first trimester. It seems plausible to me that later miscarriage or induced abortion (e.g. ~20 weeks) might be more like a previous birth where a very short IPI is thought to be associated with greater risks. I think that they need to spell out in the discussion that they were unable to sub-classify miscarriages or induced abortions by gestational age, that the vast majority will be first trimester and their results should be extrapolated cautiously to second trimester miscarriages or induced abortions. 

Comments from Reviewers:

Reviewer #1: The authors have addressed my concerns and I now recommend publication

Reviewer #3: Many thanks for taking on board my comments and suggestions and revising the manuscript accordingly. I have no further comments and feel that this paper will add to the body of evidence supporting early conception following miscarriage and induced abortion.

[LINK]

---

## [Editor Report · Decision Letter 3]

19 Oct 2022

Dear Dr Tessema, 

On behalf of my colleagues and the Academic Editor, Professor Gordon Smith, I am pleased to inform you that we have agreed to publish your manuscript "Interpregnancy interval and adverse pregnancy outcomes among pregnancies following miscarriages or induced abortions in Norway (2008-2016): a cohort study" (PMEDICINE-D-22-01509R3) in PLOS Medicine.

I noted two very minor further revisions to be made prior to publication. Please see below:

* Where you report statistics in the abstract, please insert a space between the text and the parentheses - see lines 

28, 31, 32, 33, 35, 38 

* Line 374 – suggest replacing “occurred” with “occurring” such that the sentence reads as follows: “Although the 

gestational age at previous miscarriage or induced abortion affects the IPI, we did not have information on exact 

gestational age for those miscarriages or induced abortions occurring before 12 weeks, which comprised 99% of 

miscarriages and 96% of induced abortions.”

PRESS

Sincerely, 

Philippa Dodd, MBBS MRCP PhD 

Senior Editor 

PLOS Medicine